# Development of a Cutting Technique Modification Training Program and Evaluation of its Effects on Movement Quality and Cutting Performance in Male Adolescent American Football Players

**DOI:** 10.3390/sports11090184

**Published:** 2023-09-17

**Authors:** Lutz Thieschäfer, Julius Klütz, Julian Weig, Thomas Dos’Santos, Dirk Büsch

**Affiliations:** 1Institute of Sport Science, Carl von Ossietzky University Oldenburg, 26129 Oldenburg, Germanydirk.buesch@uol.de (D.B.); 2Department of Sport and Exercise Sciences, Musculoskeletal Science and Sports Medicine Research Centre, Manchester Metropolitan University, Manchester M1 7EL, UK; t.dossantos@mmu.ac.uk; 3Manchester Institute of Sport, Manchester Metropolitan University, Manchester M1 7EL, UK

**Keywords:** agility, change of direction, injury prevention, sidestep, youth

## Abstract

This study developed a cutting technique modification training program and investigated its effects on cutting performance and movement quality in adolescent American football players. For six weeks, an intervention group (IG) of 11 players participated in 25 min cutting technique modification training sessions integrated into team training twice a week, while a control group (CG) of 11 players continued their usual team training. Movement quality was assessed by evaluating 2D high-speed videos, obtained during preplanned 45° and 90° cutting tests, using the Cutting Movement Assessment Score (CMAS) qualitative screening tool. Cutting performance was assessed based on change of direction deficit (CODD). Significant interaction effects of time × group were found for CMAS in 45° and 90° cuttings (*p* < 0.001, η_p_^2^ = 0.76, *p* < 0.001, η_p_^2^ = 0.64, respectively), with large improvements in the IG (*p* < 0.001, *g* = −2.16, *p* < 0.001, *g* = −1.78, respectively) and deteriorations in the CG for 45° cuttings (*p* = 0.002, *g* = 1.15). However, no statistically significant differences in CODD were observed pre-to-post intervention. The cutting technique modification training was effective at improving movement quality without impairing cutting performance, and it can be used by practitioners working with adolescent athletes.

## 1. Introduction

Cutting maneuvers are frequently performed in multidirectional sports, such as basketball, netball, ultimate frisbee, soccer, rugby, and other codes of football [1,2,3,4,5,6], among other change of direction (COD) actions (e.g., spins and turns) [7], to rapidly change the direction of movement in both offensive and defensive play. On the offensive, attacking players change direction to deceive/evade opponents or to receive a pass [8]; vice versa, in defensive scenarios, defenders are intent on pursuing opposing players and perform cuttings to avoid separation or to intercept passes to (re)gain ball possession [9]. Considering the high frequency of cutting maneuvers in invasion sports and their significance for decisive moments in gameplay, the development of these skills is fundamental for high sports performance [10].

Although quick changes of direction are important for athletic performance, they are also associated with noncontact anterior cruciate ligament (ACL) injuries [11]. In particular, cutting movements that involve a lateral foot plant have been identified as the most frequent mechanisms of ACL tears in basketball [12], handball [13], rugby union [14], Australian football [15], and American football [16]. Rapid cutting maneuvers can result in the potential generation of high and hazardous multiplanar knee joint loads (i.e., knee valgus and internal rotation moments, as well as anterior tibial shear), which the ACL must sustain [17]. These potentially hazardous knee joint loads are amplified when aberrant movement quality is exhibited, such as lateral trunk flexion, extended knee postures, and valgus [18]. Ligament injuries ultimately occur when the applied strain exceeds the ACL’s mechanical tolerance (i.e., supraphysiological loading) [19,20]; thus, modifying athletes’ movement quality to subsequently reduce knee joint loads could be an effective strategy to mitigate injury risk [18].

ACL injuries are not exclusively a problem in adult sports. The rising incidence of pediatric ACL tears, likely as a result of the more frequent enrollment of young athletes in organized sports and high-demand year-round competition, points to an emerging issue in junior sports as well [21,22,23,24,25]. Incidences increase significantly between the ages of 11 and 13 [23,26], with a peak during high school [21]. Long-term impairments in sports performance and participation, as well as negative effects on physical and cognitive development and health, can be concomitants of ACL injuries in children and adolescents [24]. Therefore, injury prevention strategies should target modifiable risk factors for injury to counteract this trend; although, there is not yet a clear picture of modifiable risk factors for ACL injury in the pediatric population [27]. Nonetheless, athletes’ biomechanical movement patterns were highlighted as a key modifiable risk factor for injury in the 2018 International Olympic Committee consensus statement on the prevention, diagnosis, and management of pediatric ACL injuries [28]. Consequently, the adoption of safer cutting techniques can be considered a viable prevention strategy as it reduces knee joint loading, thus mitigating injury risk potential in young athletes.

In a systematic review, Donelon et al. [18] examined the biomechanical determinants of knee joint loading during cutting maneuvers and provided a framework of technical movement characteristics associated with safer cuts (i.e., reduced knee joint loads) such as the reduction of lateral foot plant distance, minimization of knee valgus, avoidance of contralateral trunk flexion, and application of a penultimate foot contact breaking strategy. However, some of the proposed techniques and mechanics are also associated with decreased cutting performance (e.g., decreased lateral foot plant distance), highlighting that cutting performance and safe mechanics are in direct conflict [11]. This inverse relationship has been termed the “performance–injury conflict” concept [17]. Nonetheless, the performance–injury conflict does not generally apply to all kinematics that are associated with reduced injury risk potential. Some of them are largely unrelated to cutting performance (e.g., knee valgus), while other safe cutting kinematics have a positive relationship with performance (e.g., trunk rotation and lateral flexion toward the direction of travel, as well as a clear breaking strategy) [11,29,30,31,32]. It is less likely that athletes and coaches can be convinced to implement technique training programs that sacrifice performance for injury risk mitigation [8]. Likewise, from an ethical point of view, it would be reckless to train techniques that enhance performance while putting athletes’ health in jeopardy. Consequently, cutting technique modification programs, irrespective of whether performance enhancement or injury prevention is the primary goal, should mainly target movement strategies that yield dual benefits (i.e., improved performance and injury risk reduction) [11,17,33]. Furthermore, according to a survey of girls’ soccer coaches, ACL injury prevention programs are more likely to be implemented if coaches believe the program improves performance [34]. Moreover, female adolescent athletes are more likely to participate in these programs if data indicate fewer injury risk factors [35]. Therefore, concomitantly enhanced performance is a major selling point for the successful implementation of and compliance with ACL injury prevention programs that are considered effective.

Technique modification programs appear to be an effective training strategy for addressing hazardous cutting kinematics and mechanics [36,37,38]. The effectiveness of currently available ACL injury prevention strategies has been questioned as improvements in movement patterns do not transfer to the field and are not well sustained over time, thereby questioning whether these strategies are effective long-term [39,40]. Therefore, it has been suggested to make use of principles from the domain of motor learning (e.g., implementing observational practice, external focus of attention, or self-controlled practice) to enhance skill development and increase transfer and retention [39,40,41,42].

From a performance point of view, movement technique is, according to current agility models, acknowledged as a key determinant of agility performance, along with physical qualities and perceptual–cognitive factors [43,44,45,46]. It has recently been proposed that alternative types of training, including those focusing on movement techniques, may have a similar or even greater effect on cutting performance than traditional training approaches (i.e., strength/power training), despite shorter program durations [47]. Nevertheless, the role of movement technique in cutting performance and its trainability in young athletes is poorly understood [48]. This is particularly problematic since findings ascertained in adults cannot implicitly be applied to the youth population because of the consequences of growth and maturation and their impact on trainability [49,50]. In addition, cutting techniques and mechanics not only differ between adult and youth athletes [51,52], but they are even maturation dependent [53,54,55,56,57].

Research on cutting technique modification in the youth population is scarce. Nevertheless, the few studies available report promising results. Celebrini et al. [58,59] investigated the effects of a technique modification intervention on sidestep cuttings in adolescent soccer players. Improvements in biomechanical risk factors were observed after the intervention; however, performance measures were not reported. Dos’Santos et al. [60] evaluated the effects of technique modification training on cutting performance and movement quality in adolescent soccer players in comparison to the control group (CG). Large interaction effects of time (pre- and post-test) × group (intervention group (IG) and CG) were reported for cutting performance and movement quality parameters, with meaningful improvements in the IG.

To the best of our knowledge, no study has yet assessed the effects of cutting technique modification training on cutting performance and movement quality in adolescent American football athletes. Therefore, the objectives of this study were, first, to develop a 6-week technique modification training program that targets safe and performance-enhancing movement patterns, incorporating specific aspects from the motor-learning domain, and second, to evaluate its effects on cutting performance and movement quality in male adolescent American football athletes. It was hypothesized separately that the 6-week cutting technique modification intervention results in higher movement quality (Hypothesis 1) and faster cutting performance (Hypothesis 2) in an IG versus a CG performing a standard training program.

## 2. Materials and Methods

### 2.1. Experimental Design

This study was conceptualized as a nonrandomized, controlled intervention study with a pre- to post-test design to evaluate the effects of a 6-week cutting technique modification training on cutting performance and movement quality in adolescent American football players in comparison to a CG. The study period was set to begin in early 2023, during the preseason. The IG participated in cutting technique training sessions (two sessions per week) integrated into regular sport technical and tactical training, whereas a CG continued their usual team training routine. Cutting tests were performed before and after the intervention period. COD performance was assessed based on COD deficit (CODD) [61], and the Cutting Movement Assessment Score (CMAS) [62] qualitative screening tool was used for assessing movement quality.

### 2.2. Subjects

A minimum sample size of *n* = 22 was estimated a priori according to the recommendations of Beck [63] using G*Power 3.1.9.7 (Heinrich Heine Universität Düsseldorf, Düsseldorf, Germany) [64] based on the mean effect sizes of within–between interaction effects of a repeated measures ANOVA for CMAS scores observed in a comparable study [60] to achieve sufficient power (*F*-test, η_p_^2^ = 0.39, α = 0.01, 1−β = 0.80).

Thirty-two male adolescent American football players from three German 3rd-division teams playing in the same age group volunteered to participate in this study (Figure 1). The study was approved by the Research Ethics Committee of the Carl von Ossietzky University of Oldenburg (approval number: Drs.EK/2023/008), and all subjects and their legal guardians gave their prior written informed consent. None of the subjects had suffered severe injuries to their lower extremities in the past year. Initially, the IG consisted of *n* = 18 players from one team, and the CG consisted of *n* = 14 players from the other two teams. Seven players from the IG and three players from the CG had to withdraw because of injury (unrelated to the study), illness, or a resulting low attendance rate (minimum compliance for inclusion was set to 58%). The mean attendance rate was 75.0 ± 15.8% (9 ± 1.9 training sessions). Ultimately, the IG (height = 179.1 ± 7.6 cm; mass = 75.1 ± 14.7 kg; body fat = 17.2 ± 8.6%; age = 15.9 ± 1.5 years; maturity offset = 2.4 ± 1.3 years; *M* ± *SD*) and the CG (height = 178.0 ± 6.8 cm; mass = 78.8 ± 19.8 kg; body fat = 17.3 ± 9.6%; age = 17.0 ± 1.1 years; maturity offset = 2.8 ± 0.9 years) consisted of *n* = 11 players each.

### 2.3. Procedures

The pre- and post-testings were performed on the same weekday and at the same time to control for circadian rhythm effects. The subjects were allowed to drink ad libitum to ensure adequate hydration. Before testing, the subjects’ recovery status was checked using the Short Recovery and Stress Scale [65], and anthropometrical assessments were performed. Standing height and sitting height were measured with a stadiometer (Seca 217, Seca GmbH & Co. KG., Hamburg, Germany). A body composition analyzer (InBody 270, InBody Co., Ltd., Seoul, Republic of Korea) was used to measure body mass and estimate fat mass. Maturity offset was estimated with the software BIO-FiNAL 3.4 (Institute for Applied Training Science, Leipzig, Germany) [66], using the equations of Mirwald et al. [67]. Subsequently, the subjects completed a supervised ten-minute standardized warm-up program.

#### 2.3.1. Cutting Assessment (Performance and Movement Quality)

The cutting assessment was used to determine COD performance (time) of the sidestep technique [68] and to evaluate cutting movement quality (regarding ACL injury risk potential). The sidestep technique was chosen because it is one of the more common techniques in multidirectional invasion sports [2,5,69,70]. Cutting angles of 45° are probably more representative of most cutting scenarios in American football, whereas sharper cutting angles increase the risk of injury because of the increased load on the knee joint [71,72], with a 90° cutting angle posing the greatest risk [73]. Hence, both 45° and 90° cutting angles were examined. The test setup is shown in Figure 2. Example photos and videos of the cutting test are openly available in Open Science Framework at DOI 10.17605/OSF.IO/X6E3V. COD performance times were recorded using an electronic timing gate system (Witty System, Microgate, Bolzano, Italy) set up at about hip height. Three synchronized high-speed cameras (VCXU.2-50C and VCXU-50C, 100 Hz sampling frequency, Baumer Holding AG, Frauenfeld, Switzerland) captured 2D video footage of the sidestep movement in the frontal and sagittal plane (placed at 3 m and 5 m distances to the cutting position at a height of 90 cm, respectively). Additionally, a tablet (iPad Pro, 120 Hz sampling frequency, Apple Inc., Cupertino, CA, USA) was positioned at 20° to capture potentially pre-rotated cuttings on the frontal plane [62]. Two spotlights provided adequate lighting for the video recordings. The tests were performed indoors on artificial turf.

Two practice trials at 75% of the maximum perceived effort were performed prior to testing. The test was performed under preplanned and unplanned conditions, but only the preplanned condition was considered in the study. The order of the conditions was randomized to account for possible fatigue, habituation, or learning effects. The subjects were instructed to perform the test with maximum effort and to start in a two-point stance (freely selectable front leg), 50 cm behind the first gate to prevent inadvertent triggering of the timing. Subjects sprinted through the start timing gate, straight through a second timing gate at 5 m for split times (Split5m) to approximate entry speed, and onward to the cutting point at 7 m. A flat cone placed at 7.5 m was used as a marker for the latest cutting point. At the cutting point, participants had to sidestep and reaccelerate toward the exit gate at a 3 m distance as fast as possible. Trials were performed until two valid trials in each direction (left/right) and cutting angle (45°/90°) were recorded (eight trials in total). Recovery of at least one minute was provided between the trials.

Mean COD performance times for each cutting angle were calculated and used for the CODD computations. The CODD represents the additional time required to change direction compared to a pure linear sprint over an equivalent distance and, thus, provides a more isolated measure of the actual cutting performance [74]. CODD was calculated using the formula: CODD = mean cutting assessment time—mean sprint assessment time [61].

The movement quality was assessed by screening the 2D video footage of the sidestep cuts with the movement analysis software TEMPLO^®^ (Version 2022.1, Contemplas GmbH, Kempten, Germany) against the 9-item CMAS tool [62]. The CMAS is a validated [75,76] and reliable [77] field-based qualitative screening tool to identify high-risk postures associated with increased noncontact ACL injury risk during sidestep cutting. Two raters (J.K. and J.W.; sports scientists with three years of coaching experience in invasion sports and sports therapy) were assigned to independently screen the trials. The raters were allowed to watch the footage as often as desired and use the included software tools at their discretion. Before the screening, both raters participated in a 90 min training session for the CMAS screening tool led by one of its originators (T.D.S.). The awarded scores served as indicators of movement quality regarding ACL injury risk potential. Means from both rater scores were calculated for each trial. Mean CMAS scores for 45° and 90° cuttings were used for further analysis.

#### 2.3.2. Sprint Assessment

Following the cutting assessment (with at least five minutes rest), linear speed was assessed with a 10 m sprint test performed indoors on artificial turf. Photocells and reflectors (Witty System, Microgate, Bolzano, Italy) were positioned at 0 m, 5 m, and 10 m at approximately hip height to record the sprint times (Sprint10m) and 5 m split times (Sprint5m). The subjects started in a two-point stance (freely selectable front leg) 50 cm behind the first timing gate and performed three trials at maximum effort with at least one minute rest between each trial. The mean of the three trials was used for further analysis and CODD calculations.

### 2.4. Training Intervention

The usual training sessions of the IG and CG consisted of 20 min of warm-up, followed by 25 min of conditioning drills (mostly speed, agility, and quickness), and 60 min of team training. The technique modification training replaced the conditioning phase twice a week (≥48 h between sessions) for six consecutive weeks in the IG. According to the training plans, the training volumes of the IG and CG were comparable overall. Three experienced instructors (L.T., J.K., and J.W.) organized and led the technique modification training sessions. Instructors supervised subgroups with a maximum of six athletes, randomly reassigned at each session to mitigate potential instructor effects. Each technique modification training session comprised 3–4 deceleration, COD, and/or agility exercises performed outdoors on an artificial field-turf pitch with helmets on (a detailed description of the training exercises performed can be found in the Appendix A Appendix A), with the cutting tasks performed from both limbs (i.e., toward both left and right directions).

The technique modification training program aimed to foster and reinforce specific movement patterns (i.e., penultimate foot contact dominant braking strategy, trunk lean and rotation toward the direction of travel, reduction of tibia abduction angle, and active limb at touchdown) of the sidestep technique [18,32]. The desired movement patterns have been targeted in former technique modification interventions to simultaneously promote higher cutting performance and lower ACL injury risk potential [60,78,79]. A detailed model of the target sidestep technique is described in Appendix B.

Since the conducted training intervention differs from previous cutting technique modification strategies, a comprehensive description and rationale of the approach are provided. Recent findings suggest that motor learning methods of nonlinear pedagogy are superior to linear pedagogy approaches in modifying kinematic factors related to noncontact ACL injuries [80,81]. However, in the present study, because of the performance–injury conflict of certain sidestep kinematics, a linear pedagogy approach with the provision of a targeted technique was pursued, as athletes could otherwise acquire a cutting technique that is safer regarding ACL injury risk but, on the other hand, detrimental to performance. The present training approach considers recent recommendations for enhancing ACL injury prevention programs [39,40,41,42], as it utilizes several principles from the domain of motor learning. Training in small groups allowed for an appropriate athlete-to-coach ratio (approx. 6:1), time efficiency, and observational practice. Alternating between observing (during rest) and physically practicing a task has been shown to enhance skill learning, transfer, and retention [41]. The general structure of the technique training (Figure 3) was aligned with the “cutting development framework” of Dos’Santos et al. [82], which was recently adopted for multidirectional speed development in maturing athletes [83]. The framework distinguishes three phases: technique acquisition, technique retention and integrity, and movement solutions [82]. The specificity and the cognitive and physical load of the training exercises progressively increase across the phases [84]. In the present training approach, the individual phases are not strictly demarcated but rather merge smoothly into one another. Although the training program was aligned with the cutting development framework, the actual exercises were formatively developed during the intervention period to optimally scale with athletes’ training progress.

Each exercise was verbally explained and demonstrated by the instructors before being practiced. In the first phase (sessions 1–5), short (~10 s) slow-motion expert video demonstrations (J.K.) with graphical guidance on proper decelerations or sidesteps were presented on convertibles (IdeaPad Miix 310, Lenovo Ltd., Hong Kong, China) prior to each trial (video demonstration frequency = 100%) to initially introduce and teach the target technique of the sidestep model and provide model learning [86]. To avoid overloading learners with too much information, instructions accompanying the demonstrations were limited to one critical cue to focus on [87]. Video demonstrations have been shown to facilitate motor learning [85,88], transfer [89], and retention [90], especially at the early stages of learning. Almost immediately (~10 s) after the performed trials, descriptive and prescriptive verbal feedback was given in combination by the instructors to assist the athletes in understanding how to improve their technique and to develop intrinsic feedback mechanisms for error detection [91,92]. In the first sessions, the feedback frequency was set at 100% and then gradually reduced to avoid reliance and promote autonomy. Although feedback was given frequently, it was mostly provided after successful attempts by positively addressing the correct movement patterns, which has a beneficial effect on learning and retention [41,93,94,95]. Both instructions and feedback predominantly used standardized analogies or cues with an external focus of attention as recommended in the skill-training communication model [86]. Analogies and external cues have been recommended when working with young athletes [96,97] and when teaching agility skills [98,99] as they accelerate the learning process, enhance the production of effective and efficient movement patterns, and improve athletes’ performance [40,93,100]. The cues given were in accordance with current recommendations of verbal coaching cues for faster and safer cutting performance [82] and comparable to cueing practices in previous studies [60,78,79]. The physical demands were initially kept low by practicing the drills with submaximal effort (50–75% of perceived maximum effort) and with low cutting angles (45°–70°). Similarly, the cognitive load was rather low when practicing preplanned cuttings in a simplified environment (e.g., no teammate or opponent interaction). In addition, decelerations and sidesteps were trained separately at first and then together as a whole practice. The resulting reduced task difficulty allowed the athletes to focus on the correct execution of the technique. The physical load and cognitive effort were gradually increased throughout the phase as the athletes’ skill levels increased. It has been shown that learners are most adept at determining the optimal task difficulty (to facilitate learning), according to their ability to perform the task [101]. Hence, in some exercises, the athletes were given the option to self-control physical load (i.e., adjusting speed) to provide individualized and optimal task difficulty.

The following phase (sessions 5–10) focused on maintaining the newly trained technique by practice with increased task complexity (e.g., multiple cuts and inclusion of implements) and increased physical (e.g., higher velocity and cutting angles) and cognitive demands (e.g., cuts performed in response to anticipatable sport-specific stimuli). The training exercises were increasingly conducted in dyads (e.g., mirror drills or cutting past a passive opponent), which can provide additional learning advantages by increasing motivation through social interaction and competition [40,41]. Since athletes presumably already attained a rough mental representation of the target sidestep kinematics, the frequency of video demonstration was set to once per exercise. Similarly, the frequency of verbal feedback was reduced to 50% because the athletes had likely developed a good sense of when a movement execution was incorrect. In addition, video feedback was introduced in this phase to further support motor learning [85,102,103]. Tablets (iPad Pro, Apple Inc., Cupertino, USA) with a video delay app (Videoverzögerung für Sport 1.8.1, GraafICT) were set up during the training sessions to give the athletes the opportunity to receive immediate (~10 s interval length) video feedback of the performed movement [85]. At the beginning of the phase, video feedback frequency was set to 100%. As the phase progressed, the frequency and type of feedback were controlled by the athletes at their own discretion (i.e., athletes could request verbal and/or video feedback after every trial). A self-controlled feedback frequency grants the athletes a certain degree of autonomy, which has the potential to enhance learning, motivation, and retention [40,93,95,104,105,106]. The frequency of the video demonstrations was also reduced to a self-selected level by the end of the phase.

In the last phase (sessions 11 and 12), simulated sport-specific scenarios (e.g., cutting past an active defender approaching from the side) conduce to transfer and skill retention [82]. This phase aimed to provide the athletes with a random, as well as more representative, environment in which to retrieve, select, and execute appropriate movement patterns under high physical and cognitive load. The exercises were performed in full American football clothing and equipment for higher representativity. Furthermore, American football’s specific actions such as ball receiving and carrying, as well as tackling were included to increase task difficulty. The quantity of the cuttings performed was increased to provoke a high physical load. Moreover, some drills had a competitive character to maintain motivation and intensity. To vary movement preparation times and cognitive demands, cuttings were mostly performed in response to stimuli that were either anticipatable or unanticipatable.

According to the cutting development framework, the practice structure changes over the phases from block to serial to random practice to increase contextual interference [82]. However, in the present study, exercises were only performed in block practice.

### 2.5. Statistical Analyses

The CODD and CMAS at 45° (CODD45, CMAS45, respectively) and 90° cutting angles (CODD90, CMAS90, respectively) were considered as primary dependent variables. In addition, the 5 m split times of the sprint test (Sprint5m) and the cutting test (Split5m45, Split5m90) were used for further in-depth analyses. Data were statistically analyzed with IBM SPSS Statistics (29.0.0.0, IBM, Armonk, NY, USA). The within-session reliability was determined by calculating the intraclass correlation coefficient (*ICC*; model designation according to McGraw and Wong [107]: *ICC*(A,k)) and interpreted according to Koo and Li [108] as poor (<0.50), moderate (0.50–0.75), good (0.75–0.90), and excellent (>0.90). In addition, the coefficient of variation (*CV*) was calculated as $CV=\frac{SD}{M}\times 100$ for each subject and averaged across subjects, the standard error of measurement (*SEM*) was calculated as $SEM={SD}_{pooled}\sqrt{1-ICC}$ [109], and the minimum detectable change with a 90% level of confidence (*MDC*_90_) was calculated as ${MDC}_{90}=SEM\times 1.64\times \sqrt{2}$ [109]. Since responses to ACL injury prevention programs can be very dissimilar in individuals [110,111], subjects were classified into positive-, non-, and negative responders according to subjects’ pre–post differences in comparison to the respective *MDC*90.

To assess intra-rater reliability, the raters screened and graded a subset of 22 videos (one randomly selected trial from each subject) a second time at least one week apart. In addition, a third rater (T.D.S.; biomechanist with 10 years of experience and certified strength and conditioning specialist) graded the same subset, and these scores were collated against raters’ original scores to establish inter-rater reliability by a blinded assessor. *ICC*(A,k) and *ICC*(C,k) were calculated to determine the intra-rater and inter-rater reliability, respectively [107].

Data were checked for normal distribution using the Shapiro–Wilk test and examined for skewness, kurtosis, and unimodal distribution. Two-way mixed repeated ANOVAs were conducted (group × time), with group as the between-subjects factor (IG and CG), and time as the within-subjects factor (pre- and post-test) for the CODD and CMAS scores at both angles. The statistical significance level was adjusted to α = 0.01 to mitigate alpha inflation and for more conservative testing. An SPSS syntax by Wuensch [112] was applied to calculate the 90% CI for η_p_^2^. Furthermore, changes from pre- to post-test were investigated with paired t-tests in both groups, and independent t-tests were used to compare variables between groups. In addition to the classical null hypothesis significance testing of the t-tests, Bayesian analyses were performed with a Cauchy distribution prior centered on zero and a scale parameter of 1.0. An orientation of small, medium, and large effects was based on Hedges’ *g* ranges of 0.10–0.29, 0.30–0.40, and ≥0.50 and η_p_^2^ ranges of 0.010–0.059, 0.060–0.149, and ≥0.150, respectively [113]. Bayes factor interpretations were according to Lee and Wagenmakers’ classification scheme [114]. The SPSS syntax collection of Loffing [115] was used for raw data visualization (line plots for 2 × 2 mixed designs with means, confidence intervals, and medians).

## 3. Results

Anthropometric measures and age characteristics of IG and CG were comparable since independent-sample t-tests could not detect statistically significant differences between groups (*p* > 0.069). Cuttings to the left and right side were pooled as described in Section 2, because no statistically significant difference in the CMAS between cutting sides was observed when sorted regarding left/right side, preferred kicking leg, preferred jumping leg, or preferred cutting direction (*p* > 0.015).

Parameters of within-session reliability for pre- and post-tests of the IG and CG for CMAS, CODD, and Sprint10m are presented in Table 1. The *ICC*s for the CMAS showed good (*ICC* = 0.76–0.88) reliability, while the *ICC*s for the CODD were considered moderate to excellent (*ICC* = 0.74–0.91) and for Sprint10m excellent (*ICC* = 0.93–0.95). The CMAS, CODD, and Sprint10m displayed *CV* ranges of 15.61–21.00%, 13.79–39.86%, and 2.28–2.46%, respectively, with higher *CV*s in the 45° cuttings. The *SEM*s of 0.30–0.41, 0.04–0.05 s, and 0.03–0.04 s were observed for CMAS, CODD, and Sprint10m, respectively. The *MDC*90s for the CMAS, CODD, and Sprint10m were determined as 0.87–0.96, 0.09–0.11 s, and 0.07 s, respectively. Both raters demonstrated excellent intra-rater reliability (rater A: *ICC* = 0.99, 95% CI (0.97, >0.99); rater B: *ICC* = 0.91, 95% CI (0.74, 96)), while the inter-rater reliability was considered moderate (*ICC* = 0.71, 95% CI (0.42, 0.87)).

Pre- and post-test descriptives of the primary outcomes, split times, their differences, and individual responses are presented in Table 2. Individual changes and changes in group means of the CMAS and CODD are illustrated in Figure 4 and Figure 5. The data were normally distributed to allow for parametric procedures to be used. The movement quality of the athletes was meaningfully affected by the technique modification intervention, as the mixed ANOVA revealed statistically significant large interaction effects for group × time for CMAS45, *F*(1, 20) = 63.77, *p* < 0.001, η_p_^2^ = 0.76, 90% CI (0.56, 0.84) and CMAS90, *F*(1, 20) = 36.11, *p* < 0.001, η_p_^2^ = 0.64, 90% CI (0.38, 0.75). Statistically significant differences between pre- and post-test in movement quality were observed, with improvements (i.e., reductions) in CMAS45 and CMAS90 in the IG, *t*(10) = −7.75, *p* < 0.001, *g* = −2.16, 95% CI (−3.22, −1.07), BF_10_ = 1818.22, *t*(10) = −6.41, *p* < 0.001, *g* = −1.78, 95% CI (−2.71, −0.83), BF_10_ = 432.53, respectively, and deteriorations (i.e., increases) in CMAS45 in the CG, *t*(10) = 4.12, *p* = 0.002, *g* = 1.15, 95% CI (0.39, 1.87), BF_10_ = 23.37. The resulting Bayes factors indicate extreme evidence in favor of t-tests’ alternative hypothesis for IG data (i.e., improvements) and strong evidence for CG data (i.e., deteriorations). The intervention was slightly more effective in improving CMAS45 compared to CMAS90, as shown by the effect sizes obtained (*g* = −2.16 vs. *g* = −1.78). Pre-to-post-test differences for CMAS45 and CMAS90 were statistically significantly different between groups, *t*(20) = −7.99, *p* < 0.001, *g* = −3.28, 95% CI (−4.55, −1.97), BF_10_ = 99,793.62, *t*(20) = −6.01, *p* < 0.001, *g* = −2.47, 95% CI (−3.55, −1.34), BF_10_ = 2504.59, respectively.

Two-way mixed ANOVA showed no statistically significant interaction effects for group × time, *F*(1, 20) = 0.15, *p* = 0.701, η_p_^2^ = 0.01, 1−β = 0.02, for CODD45. Likewise, no statistically significant main effects for group, *F*(1, 20) = 6.47, *p* = 0.019, η_p_^2^ = 0.25, 1−β = 0.41 or time, *F*(1, 20) = 1.66, *p* = 0.213, η_p_^2^ = 0.08, 1−β = 0.08 for CODD45 were observed. Comparable results were obtained for CODD90, with no statistically significant interaction effects for group × time, *F*(1, 20) = 2.66, *p* = 0.119, η_p_^2^ = 0.12, 1−β = 0.14 or main effects for group or time, *F*(1, 20) = 8.16, *p* = 0.010, η_p_^2^ = 0.29, 1−β = 0.52, *F*(1, 20) = 0.07, *p* = 0.800, η_p_^2^ < 0.01, 1−β = 0.01, respectively. Thus, no meaningful effects of the cutting modification intervention on cutting performance were detected.

No statistically significant interaction effects for group × time or main effects for group or time for Sprint10m were observed: *F*(1, 20) = 1.77, *p* = 0.198, η_p_^2^ = 0.08, 1−β = 0.09; *F*(1, 20) = 0.98, *p* = 0.334, η_p_^2^ = 0.05, 1−β = 0.05; and *F*(1, 20) = 4.28, *p* = 0.052, η_p_^2^ = 0.18, 1−β = 0.25, respectively.

Differences between the split times of the sprint and cutting test were found: During the pre-test, the paired-sample t-tests revealed statistically significant differences between Sprint5m and Split5m45, *t*(10) = −3.21, *p* = 0.009, *g* = −0.89, 95% CI (−1.55, −0.21), BF_10_ = 6.30, and Split5m90, *t*(10) = −6.49, *p* < 0.001, *g* = −1.81, 95% CI (−2.74, −0.84), BF_10_ = 474.49, in the CG but not in the IG (*p* > 0.115). These differences in the CG were not observed post-intervention (*p* > 0.532), whereas the Sprint5m times were significantly faster than the split times observed in the 45° and 90° cuts in the IG, *t*(10) = −4.69, *p* < 0.001, *g* = −1.30, 95% CI (−2.07, −0.51), BF_10_ = 51.03; *t*(10) = −3.95, *p* = 0.003, *g* = −1.10, 95% CI (−1.81, −0.36), BF_10_ = 18.44, respectively.

## 4. Discussion

Cutting kinematics are closely related to both cutting performance and ACL loading [11,17,32,116,117]. The aims of the present study were twofold: First, to develop a cutting technique modification training program that targets specific kinematics associated with improved performance and reduced injury risk potential while considering various principles from the motor-learning domain, and, second, to investigate its effects on cutting performance and movement quality in adolescent American football players. It was hypothesized separately that the intervention results in better movement quality (Hypothesis 1) and faster cutting performance (Hypothesis 2) in the IG when compared to a CG. The primary finding was that the technique modification approach significantly improved athletes’ sidestep cuttings movement quality without impairing performance. Avoiding “high-risk” postures during cuttings can reduce ACL injury risk potential [18]. As a result, this cutting technique modification training approach can be considered an effective program for ACL injury risk mitigation. Since improvements in cutting movement quality were only observed in the IG, Hypothesis 1 can be confirmed. However, the improved cutting technique did not translate acutely to meaningful faster cutting performance as assumed a priori, and, thus, Hypothesis 2 must be rejected.

Previous technique modification approaches to optimizing sidestep cutting techniques have produced mixed findings [58,59,60,78,79,118,119,120]. The approach of Dempsey et al. [118] aimed at altering lateral foot plant distance, foot progression angle, lateral trunk flexion, and trunk rotation in nine invasion sports athletes. Twelve training sessions utilizing external cueing, verbal and video feedback, and reference videos resulted in reduced lateral foot plant distances and more upright torso postures, accompanied by significant reductions in knee valgus moments during 45° cuttings. Although kinematic changes were evident, performance was not significantly affected, which is consistent with present findings. However, due to the missing control group, the results of Dempsey et al. [118] must be interpreted with caution. Donnelly et al. [119] pursued a combined approach of technique modification and balance training. After 28 weeks of training, no significant changes in knee kinematics or performance in 45° cuttings were observed in the 14 subjects of the IG, which was attributed to an extremely low compliance rate (45%) and a high athlete-to-coach ratio (40:1) during training sessions. In the present study, the mean training attendance rate was higher (75%), and the athlete-to-coach ratio was considerably lower (6:1), which might have facilitated more effective training sessions.

Olivares-Jabalera et al. [120] studied the effects of a 6-week technique modification program comprising a combination of landing, plyometrics, and COD exercises on cutting and landing movement quality in adult soccer players. Individual feedback provided to players was primarily comprised of external coaching cues that were comparable to cues provided in the current study. The IG significantly enhanced their movement quality during 70° cutting in comparison to a CG. Moderate to large improvements (*g* = 0.55–1.22) in the IG were reported with mean reductions in CMASs of 13.7–22.1%, comparable to the CMAS reductions of 25.9–35.2% which were observed in the current study. This finding could be attributed to a lower athlete-to-coach ratio (6:1 vs. 15:1), the use of video instruction and video feedback techniques, and the training focus being solely on sidestep cutting technique. Interestingly, in both studies, the improved cutting technique did not translate to enhanced cutting performance. Contrasting results were observed in a study by Dos’Santos et al. [78,79]. An IG of 15 male adults from multidirectional sports significantly improved their 45° and 90° cutting performance after 6 weeks of a COD speed and technique modification program in comparison to a CG. Increases in velocity at key instances of the COD were primarily responsible for performance improvements. However, multiplanar knee joint loads were not meaningfully altered post-intervention, albeit large individual differences were observed (athletes with initially high knee joint loads reduced knee joint loading), which emphasizes the importance of case-by-case analyses when assessing the effectiveness of technique modification programs [111]. Some cutting kinematics (i.e., increased knee abduction and foot progression angle and lower knee flexion range of motion) were even negatively altered indicating the adoption of a less favorable technique in the IG. The CG showed increased knee abduction angles and moments during 45° cuttings, which is in line with observed deteriorations of CG’s movement quality in the present study.

To the best of our knowledge, Celebrini et al. [58,59] and Dos’Santos et al. [60] were the only cutting technique modification studies involving youth athletes. In the training approach of Celebrini et al. [59], increased peak knee abduction angles and decreased peak knee abduction moments were observed in most of the U16 female soccer players studied after the 4-week intervention. However, the lack of a CG and a small sample size (*n* = 7) limited the results’ interpretation. The same approach was applied in a 6-week intervention in a comparable sample by Celebrini et al. [58]. In this study, a slightly larger IG (*n* = 10) and CG were used. A statistically significant group × time interaction effect for peak knee flexion angle in 55° cuttings was found, with an increase observed only in the IG, indicating the adoption of a safer movement strategy. Unfortunately, both studies did not report performance measures which reduces the likelihood of athlete and coach adoption. Given the potential performance–injury conflict [11], a holistic assessment of both, mechanics and performance is recommended [36]. Only one study could be identified that investigated performance and movement quality in a youth population [60]. Eight male U17 soccer players attended a technique modification program that was comparable to the approach used in the aforementioned study [78,79], while a CG continued their regular field-based warm-ups. Large interaction effects for Time × Group were observed for CMASs in 70° cuttings. The IG significantly improved their movement quality with reductions in CMASs of 22.5–33.9%, which is consistent with the present results. Whereas contrary findings were reported for cutting performance, enhancements in CMASs were accompanied by significantly faster CODDs in the IG in contrast to the present study. However, these improvements cannot entirely be attributed to the intervention, as faster CODDs were also observed in the CG, albeit to a smaller extent compared with the IG. The discrepancy between studies’ cutting performance results might originate from differences in training volumes. Although the training volumes of both interventions were comparable, the volumes of regular training were considerably higher in the study by Dos’Santos et al. [60]. Two strength training sessions and five technical and tactical soccer sessions per week potentially had a synergistic effect on the cutting technique intervention, since athletes had the chance to enhance their strength capacities and physically practice and exploit the biomechanical advantages of their newly acquired technique [121,122].

The causes for the stagnant COD performance observed in the present study could be unraveled by analyzing the split times. The Split5m times at both cutting angles (i.e., 45° and 90°) revealed that the IG entered the cuts, on average, at a lower speed in the post-test (1.28 s vs. 1.35 s, 1.31 s vs. 1.39 s, respectively) while the CG showed, on average, higher entry speed (1.24 s vs. 1.21 s, 1.27 s vs. 1.24 s, respectively). Sprint5m and Split5m were compared to determine possible discrepancies. During the pre-test, no statistically significant differences between Sprint5m and Split5m in the IG were observed, indicating that athletes entered the cuttings at or near their maximum speed. However, post-intervention, the IG entered the cuttings not at full speed as Sprint5m times (1.19 s) were significantly faster than Split5m times observed in 45° (1.35 s) and 90° cuttings (1.39 s). This suggests that the IG entered the cuttings post-intervention at submaximal speeds, possibly to focus on executing the targeted cutting technique as accurately as possible (i.e., speed-accuracy trade-off), which is reflected in improved CMASs. Even though a “cautious” approach provides a potentially protective impact, it should not be considered a feasible ACL injury prevention strategy, as it likely has negative performance implications [123,124]. This would also explain the lack of improvement in CODD, although individual responses are quite variable (see Table 2). It has been shown that increased velocity at key instances of the penultimate foot contact and final foot contact is largely associated with cutting performance [79,116]. Thus, the underpinning idea is to develop a pronounced deceleration ability to brake effectively (i.e., apply an adequate horizontal braking force to reduce whole-body momentum to an optimal level for the subsequent cutting movement) [31,125,126] and rapidly over the shortest possible distance/time (i.e., with very few braking steps/short braking contact time) [127] prior to direction change to maintain a high approach speed for as long as possible [32]. Effective braking was also trained in the intervention; however, during testing, some athletes occasionally fell into shuffle-step movement patterns in the transition from approach to cutting, which are associated with lower velocity maintenance [8]. Interestingly, the findings of the Split5m times in the IG are opposed to those found in the CG. Here, the Sprint5m times were significantly faster than the Split5m times in the 45° and 90° cuttings in the pre-test, while no statistically significant differences were observed in the post-test. Thus, athletes of the CG chose a more “cautious” approach strategy during the pre-test and an “all-in” approach strategy during the post-test. It has been shown that a higher approaching speed in cut maneuvers was associated with a higher occurrence of unsafe movement patterns [124], which explains the observed deteriorations from pre- to post-test in CMAS45 of the CG.

From a motor learning perspective, the stagnant COD performance observed in the IG is not a surprising phenomenon, since plateaus and regressions in performance are often seen during the skill acquisition process [92]. Actual progression is, therefore, more likely to be apparent when movement learning is viewed over a longer timescale. Nevertheless, athletes in the IG already adopted an improved movement technique since improvements in movement quality were evident. However, these improvements did not yet translate into improved COD performance. It could be assumed that with further practice of cutting movements, dynamic temporal–spatial interactions between body segments or already established functional synergies will be optimized (i.e., exploiting), ultimately resulting in movement economization and performance enhancement [128]. Although an average participation rate of nine training sessions elicited improvements in cutting quality, longer program durations are recommended, because they facilitate long-term changes in movement control [129]. Nevertheless, it would be desirable to examine retention and overall skill development in future studies to determine the optimal program duration or intervals for a refresher [10]. As an anecdotal side note, from approximately the fourth training session, some athletes occasionally expressed their displeasure after performing a cut. When asked the reasons, they stated that they were not satisfied with their execution of the movement technique. This indicated that they had already acquired a nuanced mental representation of the reference cutting technique and were able to use task-intrinsic feedback mechanisms to contrast the movement technique they had just performed with the target cutting technique. This appears to be an appropriate entry point for video feedback training, as from then on athletes can contrast their actual movement shown in the video with their mental reference technique.

Several key factors for effective cutting technique change can be deduced by combining what is known from the literature and current results. Augmented feedback seems to facilitate motor learning when modifying cutting technique and should be provided by trained professionals [130]. The modalities of the feedback (e.g., prescriptive and descriptive verbal feedback, external focus of attention, and feedback frequency) are far from trivial in this regard and should be considered carefully when designing a technique training program [40,92,94,131]. Especially, video feedback and video instruction appear to amplify motor learning of cutting movements [42,89,102,103,118]. Low athlete-to-coach ratios make it easier for coaches to provide high-quality feedback, whereas high ratios can compromise the success of a technique modification program [119]. In addition to augmented feedback, observational and self-controlled practice are factors that have been shown to enhance the learning of motor skills [40,41,105,106].

Most studies on cutting technique modification have been conducted in adult athletes, but studies’ results are not necessarily transferable between adult and youth athletes [132]. It is potentially easier to elicit changes in cutting technique in youth athletes because movement patterns of cuttings are probably not yet as stable (i.e., neural plasticity) and automatized as in adults, who likely have a higher training age (i.e., sport-specific movement experience). From a motor learning perspective, technique modifications in experienced athletes involve the change of well-established movement patterns, which can be accompanied by undesirable side effects, such as performance stagnation and a familiarization phase of individually varying duration [133,134]. This might explain the individual responses to technique modification training observed in some studies [78,79]. The processes involved in changing stable movement patterns differ from learning a new skill, and, thus, appropriate approaches for experienced athletes should be considered when designing technique modification programs to successfully modify already existing skills and to mitigate interference [133].

Early development of a safe and effective cutting technique from scratch is worth pursuing instead of practicing and automatizing an unfavorable movement technique over the years that becomes hard to correct [28]. A potential ‘window of opportunity’ for enhanced ACL injury risk reduction has been suggested for female athletes [135]. It might be in early adolescence before the onset of deficits in sensorimotor functions, which are observed in some adolescents, and peak knee injury incidence in female athletes [130,135,136]. Studies of prevention training programs to modify movement technique conducted with athletes aged 8 to 14 years reported improved movement biomechanics post intervention, with higher efficacy of programs conducted in teens [137,138,139]. It should be noted that, before applying the present intervention to prepubescents, it must be appropriately adapted to athletes’ cognitive and neuromuscular development levels [130].

However, technique training should not be the only pillar to work on in the construction of young athletes’ cutting performance and injury resilience, since other types of training, such as maximum strength, reactive strength, power, and balance training, can be effective synergists for both. Therefore, a holistic multifaceted approach is recommended [130,132,140,141]. In addition to modifications of cutting/landing techniques, improving perception and anticipation, as well as the development of physical capacities, have been highlighted as effective countermeasures of ACL injury that address athletes’ modifiable risk factors [37]. Perceptual–cognitive factors (e.g., information processing speed and pattern recognition) are trainable in youth [48] and can improve both, agility performance [43,142] and coordination of movement technique because of increased movement preparation times, which translates into reduced ACL strain [37,143,144,145]. Effective ACL prevention programs in the pediatric population include, among others, components of maximum strength, reactive strength, and neuromuscular training to enhance lower extremity muscular support [28,140,141,146]. These types of training have also been shown to enhance COD performance in young athletes [132,147,148]. Furthermore, development of physical capacities can reinforce technique change by creating the physical prerequisites for adopting desired postures and movement techniques [7,149].

### Limitations

The present study has some limitations that must be acknowledged. First, this study was chosen as a nonrandomized design. Conducting training intervention studies in sports teams in situ often poses organizational challenges, making a nonrandomized study design frequently the only feasible approach. Nevertheless, the application of randomized or crossover designs in future studies is generally endorsed. Second, although the CMAS screening tool has generally proven to be a valid and reliable method to estimate peak knee abduction moments [75,150], potential rater effects cannot be entirely ruled out. Therefore, intra- and inter-rater reliability were assessed to check for ratings’ credibility. Decent intra- (*ICC* = 0.91–0.99) and inter-rater (*ICC* = 0.71) reliability were achieved, with *ICC*s comparable to (*ICC* = 0.95 and *ICC* = 0.69, respectively) [75] or exceeding (*ICC* = 0.70 and *ICC* = 0.58, respectively) [77] those observed in other studies. As a side note, the within-session reliability of the CMASs was comparable to those observed in other studies [60,120,151]. Third, the sidestep cuttings were exclusively assessed in a preplanned COD scenario without the necessity to respond to a stimulus. Pre-planned COD scenarios occur in American football (e.g., a wide receiver running a slant route), albeit to a much lesser extent than unplanned COD scenarios in which athletes must continuously respond to opponents, teammates, and the ball [79,152]. The time to plan and prepare for an appropriate movement response is scarce when the movement is performed in reaction to or in anticipation of a stimulus compared with movements planned in advance [38,69,143,153,154]. As a result, divergent cutting kinematics and mechanics have been observed between preplanned and unplanned movement scenarios [37,69,151,155,156]. Noncontact ACL injuries were often sustained during unplanned sidestepping maneuvers [37,157], which might be explained by higher knee joint loads occurring during unplanned conditions [38,153]. Thus, tests under unplanned conditions are likely to be more representative of most sporting and injury situations where the athlete has tight time constraints when responding to a stimulus. Furthermore, the cutting kinematics obtained using on-field and in-lab environments might differ [158,159]. Therefore, when evaluating the effectiveness of a technique modification program, it is recommended that both preplanned and unplanned sidestepping tasks are performed, preferably in a representative on-field environment [119,158,159]. Nevertheless, preplanned COD movements are the mechanical and physical underpinning for agility performance (i.e., action capacity), and therefore its improvement should at least partially transfer to enhanced agility performance [46,79,160]. Furthermore, the whole continuum of movement preparation times was considered in the exercises used in this intervention, with the implementation of preplanned COD scenarios, as well as cuttings performed in anticipation (e.g., body movement) and reaction (e.g., verbal stimulus) to a stimulus. Fourth, even though the technique modification intervention has been effective, it may exceed the resources available in the training environment of the common ruck [119], questioning its general feasibility. To overcome this, technique modification training could be conducted gradually with small groups of athletes throughout the season, rather than with the entire team at once. Fifth, considering that cutting kinematics are task (e.g., cutting angle), age and sex dependent, the generalizability of the results to other cutting maneuvers or groups might be limited [52,53,55,57,72,73,161] and, thus, this serves as a future direction of research.

## 5. Conclusions

The present study developed an innovative cutting technique modification training program incorporating various principles from the domain of motor learning. Its effectiveness was investigated in a 6-week intervention conducted in adolescent American football players. The cutting technique modification training program effectively improved the sidestep cutting technique, resulting in safer movement execution without compromising performance. A high-quality movement technique is the foundation for the further development of cutting performance. Consequently, an enhanced cutting performance might be expected in the further course of agility development as athletes increasingly exploit the biomechanical advantages of their improved cutting technique.

The developed cutting technique modification training program can be used by practitioners, implemented in training regimes from early adolescence, and tailored according to athletes’ maturation. Conditioning exercises like maximum strength, reactive strength, and neuromuscular training are helpful supplements to technique modification training.

## Figures and Tables

**Figure 1 sports-11-00184-f001:**
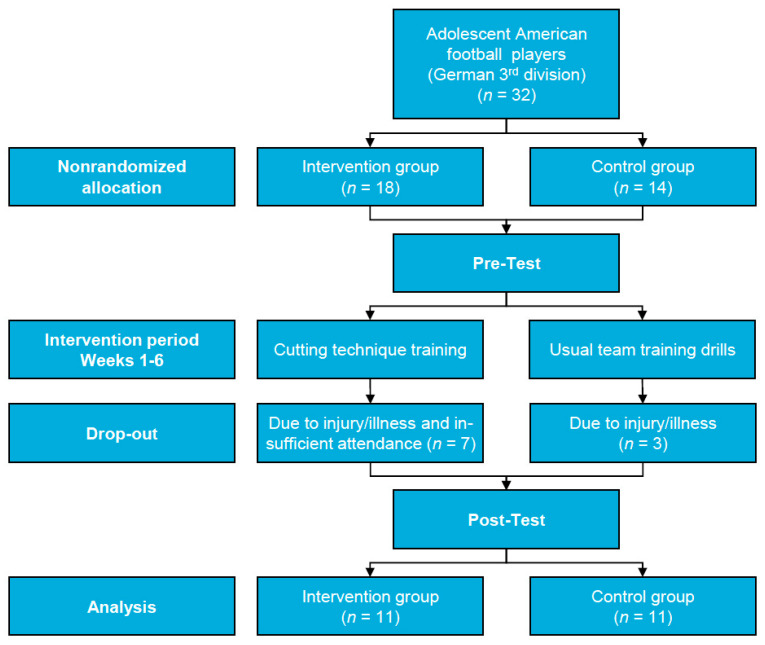
Flow chart of the study.

**Figure 2 sports-11-00184-f002:**
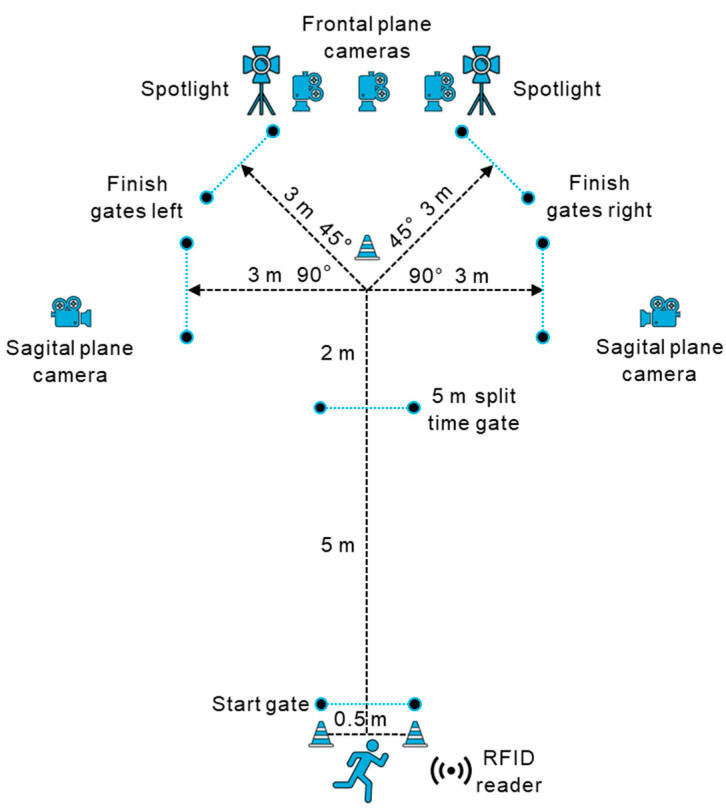
Setup of the 45° and 90° cutting tests to the left and right sides.

**Figure 3 sports-11-00184-f003:**
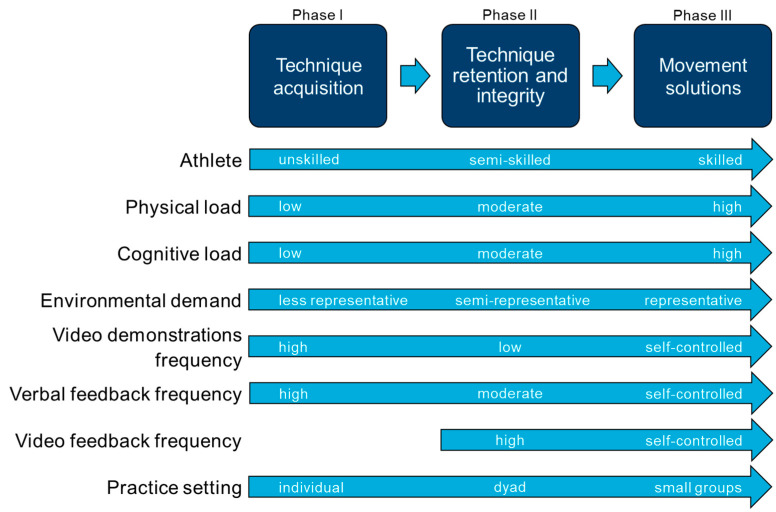
Adapted cutting development framework based on Dos’Santos et al. [82], Farrow and Robertson [84], and Nowoisky et al. [85].

**Figure 4 sports-11-00184-f004:**
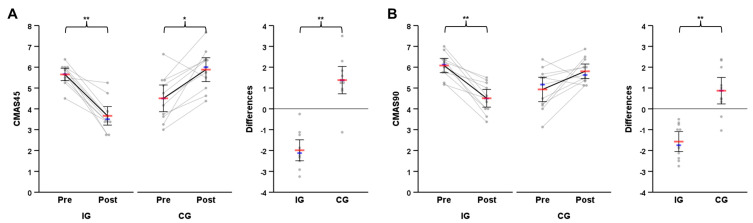
Individual pre-to-post-test changes for (**A**) CMAS45 and (**B**) CMAS90. * *p* < 0.01; ** *p* < 0.001; Red bar: mean; Blue bar: median.

**Figure 5 sports-11-00184-f005:**
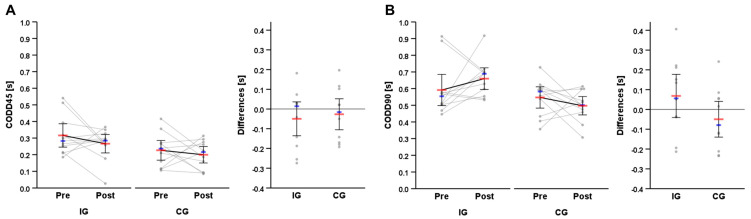
Individual pre-to-post-test changes for (**A**) CODD45 and (**B**) CODD90. Red bar: mean; Blue bar: median.

**Table 1 sports-11-00184-t001:** Within-session reliability of the pre- and post-tests for the CMAS, CODD, and Sprint10m.

Variable	Pre					Post			
	*ICC*	95% CI	*CV* (%)	*SEM*	*MDC*90	*ICC*	95% CI	*CV* (%)	*SEM*
CMAS45	0.76	(0.54, 0.89)	20.14	0.41	0.96	0.88	(0.77, 0.95)	21.00	0.30
CMAS90	0.79	(0.60, 0.90)	15.91	0.37	0.87	0.79	(0.61, 0.90)	15.61	0.30
CODD45	0.82	(0.66, 0.92)	34.29	0.05	0.11	0.74	(0.50, 0.88)	39.86	0.05
CODD90	0.91	(0.82, 0.96)	14.09	0.04	0.09	0.88	(0.77, 0.95)	13.79	0.04
Sprint10m	0.95	(0.73, 0.98)	2.28	0.03	0.07	0.93	(0.86, 0.97)	2.46	0.04

*Note*: CI: confidence interval; CMAS: Cutting Movement Assessment Score; CODD: change of direction deficit; *CV*: coefficient of variation; *ICC*: intraclass correlation coefficient; *MDC*: minimal detectable change; *SEM*: standard error of measurement.

**Table 2 sports-11-00184-t002:** Pre-to-post-test changes and athletes’ individual responses for CMAS, CODD, sprint, and split times.

Variable	Group	Pre		Post		Pre–Post Difference								IR
		*M*	*SD*	*M*	*SD*	Δ*M*	Δ*SD*	95% CI	Δ*M* (%)	*p*	*g*	95% CI	BF_10_	
CMAS45	IG	5.65	0.50	3.66	0.75	−1.99	0.85	(−2.56, −1.42)	−35.19	<0.001	−2.16	(−3.22, −1.07)	1,818.22	10-1-0
	CG	4.50	1.08	5.88	0.97	1.38	1.11	(0.63, 2.12)	30.61	0.002	1.15	(0.39, 1.87)	23.37	1-2-8
CMAS90	IG	6.08	0.58	4.51	0.72	−1.57	0.81	(−2.12, −1.03)	−25.86	<0.001	−1.78	(−2.71, −0.83)	432.53	8-3-0
	CG	4.93	0.99	5.80	0.59	0.87	1.08	(0.15, 1.59)	17.67	0.023	0.75	(0.10, 1.37)	2.91	1-4-6
CODD45 (s)	IG	0.32	0.12	0.27	0.09	−0.05	0.15	(−0.15, 0.05)	−15.76	0.282	−0.32	(−0.87, 0.25)	0.40	4-6-1
	CG	0.23	0.10	0.20	0.08	−0.03	0.13	(−0.12, 0.06)	−11.78	0.521	−0.19	(−0.73, 0.37)	0.27	4-5-2
CODD90 (s)	IG	0.59	0.16	0.66	0.11	0.07	0.19	(−0.06, 0.19)	11.53	0.248	0.34	(−0.23, 0.90)	0.43	2-4-5
	CG	0.55	0.11	0.50	0.09	−0.05	0.15	(−0.15, 0.05)	−9.07	0.307	−0.30	(−0.85, 0.27)	0.38	5-5-1
Sprint10m (s)	IG	2.04	0.16	2.10	0.15	0.06	0.09	(0.00, 0.12)	2.94	0.053	0.61	(−0.01, 1.20)	1.43	1-4-6
	CG	2.01	0.12	2.02	0.12	0.01	0.07	(−0.04, 0.06)	0.65	0.570	0.16	(−0.39, 0.71)	0.26	1-7-2
Sprint5m (s)	IG	1.25	0.13	1.19	0.16	−0.06	0.13	(−0.14, 0.03)	−4.54	0.163	−0.42	(−0.98, 0.17)	0.59	5-5-1
	CG	1.18	0.10	1.22	0.12	0.04	0.06	(0.00, 0.09)	3.56	0.054	0.61	(−0.01, 1.20)	1.41	0-7-4
Split5m45 (s)	IG	1.28	0.13	1.35	0.08	0.07	0.09	(0.01, 0.13)	5.54	0.029	0.71	(0.07, 1.32)	2.39	1-3-7
	CG	1.24	0.10	1.21	0.05	−0.03	0.07	(−0.08, 0.01)	−2.69	0.141	−0.45	(−1.01, 0.14)	0.66	2-9-0
Split5m90 (s)	IG	1.31	0.11	1.39	0.11	0.08	0.13	(−0.01, 0.16)	5.78	0.076	0.55	(−0.06, 1.13)	1.07	1-4-6
	CG	1.27	0.08	1.24	0.06	−0.03	0.06	(−0.06, 0.01)	−2.04	0.166	−0.42	(−0.98, 0.17)	0.58	2-8-1

*Note*. BF_10_: Bayes factor; CG: control group; CI: confidence interval; CMAS: Cutting Movement Assessment Score; CODD: change of direction deficit; IG: intervention group; IR: individual response, number of positive-, non-, and negative-responders (subjects’ individual response in comparison with respective *MDC*90).

## Data Availability

The datasets generated and/or analyzed during the current study are available from the corresponding author upon reasonable request.

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
