# Peer review of "Development of a Cutting Technique Modification Training Program and Evaluation of its Effects on Movement Quality and Cutting Performance in Male Adolescent American Football Players"

_sports, 2023, doi:10.3390/sports11090184_

Round 1

Reviewer 1 Report

Dear Authors,

I appreciate the opportunity to review your remarkable non-randomized controlled intervention study, employing a pre-to-posttest design to develop a 6-week technique modification training program. This program targets both safe and performance-enhancing movement patterns by incorporating specific aspects from the motor-learning domain. The study aims to evaluate the effects of this program on cutting performance and movement quality in male adolescent American football athletes.

The Manuscript ID “sports-2586579” aligns seamlessly with the thematic focus of the Special Issue on “Advances in Lower Extremity Biomechanics and Lower Extremity Injury Risk”. Furthermore, the manuscript demonstrates significant potential in terms of originality and practical application, making it a valuable addition to the proposed Special Issue.

Commendably, the researchers have exhibited a commitment to methodological quality, despite the presence of certain design limitations. Notably, the manuscript stands out for the substantial number of well-incorporated references and the meticulous quantification of participants. The comprehensive approach to the applied tests and the rigor applied in the statistical analysis also deserve recognition. Additionally, Appendix A and the Supplementary Material, providing a detailed description of the training exercises, contribute substantially to the manuscript's quality.

Nonetheless, there are areas that merit attention and will be elucidated below, with the aim of guiding you in enhancing both the impact and clarity of your manuscript:

1. Minor to moderate editing of the English language is necessary. The presence of minor typos and spelling errors may hinder readability.

2. Building upon the previous point, the presentation of the results requires consideration. Given the methodological rigor employed in the statistical analysis and the application of various statistical methods, the data presentation could be perceived as intricate. Streamlining this section for improved readability is advised, making the understanding of complex results less challenging.

3. It has come to my attention that certain figures in the manuscript employ different lettering than what is used in the main text. Moreover, I recommend verifying the resolution of the figures, as some appear to be of suboptimal quality.

4. The formatting of tables necessitates attention as well. For instance, Table 1 features bold formatting for the entire heading line, a practice not mirrored in Table 2. It is advisable to ensure consistency in the formatting of tables, including the appropriate use of bold text and accurate display of abbreviations.

5. The Conclusion section requires further refinement. The function of this section is to encapsulate the principal outcomes of your research and underscore their significance for the field of study. While reviewing the crucial findings and connections is crucial, the presence of numerous references within the conclusion appears unconventional. To this end, I encourage you to consider incorporating the content between lines 693 and 705 into the discussion. Moreover, an enhanced version of the conclusion, free from references, is recommended to summarize the study’s implications succinctly and accurately.

Please recognize that the constructive insights offered in this review are intended to serve as a roadmap for enhancing both the impact and clarity of your manuscript. By addressing these suggestions, you will undoubtedly unlock the manuscript's full potential. I eagerly anticipate your revisions.

Minor to moderate editing of the English language is required for this paper.

Reviewer 2 Report

Concerning the manuscript: Development of a Cutting Technique Modification Training Program and Evaluation of its Effects on Movement Quality and Cutting Performance in Male Adolescent American Football Players, submitted to Sports. It is a good work with sound methodology and interesting results. Overall, the paper will contribute to knowledge. With the utmost respect, allow me to give you a few suggestions.

·  Abstract cover the main aspect of the work, however a brief explanation about the role of cutting technique could be more highlighted in the abstract of manuscript.

·  The introduction topic is relevant, but the length seems very extensive.

·  The experimental design seems proper, but the authors should report some pictures demonstrating the conduction of experiments. This could be added in a supplementary file.

·  It would be interesting to determine the accumulated load (INTENSITY x VOLUME) over the entire 6-wk period. Groups should have similar training loads. Didactics would improve with the inclusion a figure detailing training load over weeks for both groups (t test in this case).

·  There is mention (in line 185) that eight trials in total were used, but what was the recovery time ?

·  I suggest the inclusion of Pearson’s correlations for understanding interactions (for example CODD vs sprint). This could be made using data from all 22 individuals (for amplifying the sample amount). If using figures, I suggest that you accurately discriminate the groups using different symbols or colors. For example:

IG group > blue square  

CG group > red circle

Reviewer 3 Report

The topic of the research is interesting and justified, since rapid changes of direction in team games can affect the risk of injury on the one hand and the effectiveness on the other.

The authors considered injury prevention (ACL) important in their research, assuming that intervention to improve the technique of changing direction would result in improved quality and performance.

The manuscript is 25 pages long, which is unusual. Reviewing the scope and structure of the publication, it can be concluded that the authors made very meticulous, almost comprehensive descriptions and explanations. The 163 literary references confirm this opinion.

The planning of work is appropriate, painstaking. Unfortunately, the already small number of elements decreased to 11 in both groups (IG, CG) by the end of the intervention period. This number of items is a serious limitation. This feeling is reinforced looking at the otherwise correct statistical apparatus.

The proofreader expects answers to the following questions:

1.           In what period did it happen The special training program (competition period, etc.)?

2.           The development lasting 6 weeks took a total of 5 hours, which raises the problem of the learning phase, the possibility of modification, because we have to assume that it is a correction of stable movement patterns.

3.           The above concern is reinforced by the results obtained, since, if it can be interpreted correctly, the effect of the intervention could be only verified in two cases.

4.           It would help the reader if, between the conclusions, yes and no answers were given to the two hypotheses, with the often theoretical explanations behind them.

The work is a valuable research initiative with a strong limitation. It should be considered that, due to the clear methodology. If the authors see the possibility, reducing the length of the paper could increase the value of the manuscript.
